# Practices of tablet splitting and dose uniformity of fragments at public hospitals in Ethiopia: A cross-sectional study supported by experimental findings

**Asmamaw Emagn Kasahun**[1,2], **Nisha Mary Joseph**[1], **Anteneh Belete**[1] *

**1** Department of Pharmaceutics and Social Pharmacy, School of Pharmacy, College of Health Sciences, Addis Ababa University, Addis Ababa, Ethiopia, **2** Department of Pharmaceutics and Social Pharmacy, School of Pharmacy, College of Medicine and Health Sciences, University of Gondar, Gondar, Ethiopia

* anteneh.belete@aau.edu.et

## Abstract

### Objective

To assess the practices and dose uniformity of tablet splitting at selected public hospitals in Northwest Ethiopia.

### Methods

A hybrid study method was employed to see the overall practices of tablet splitting. A prospective cross-sectional study was conducted to explore the practices of tablet splitting by administering structured questionnaires to patients and pharmacy professionals. Experimental data on dose and content uniformity of split tablets were obtained from the results of drugs split by study subjects. The content uniformity assay was performed using UV/Vis spectrophotometry.

### Results

A total of 241 patients and 82 pharmacy professionals participated in the cross-sectional study. The majority of patient participants (51.3%) faced problems while splitting their tablet medications and this had a significant association with the education level of the patients ($\chi2 = 60.5$; $p = 0.001$). Enteric-coated formulations were dispensed to be split, despite the precaution given by the manufacturers against splitting or crushing these products. Splitting of enteric-coated products accounts for 11% of the total drugs that were dispensed to be taken after a split. The mean of weight variation test for the half tablets does not meet the specifications set in pharmacopoeias when splitting was done by patients. The unscored haloperidol tablets were hard to split and resulted in a significant weight variation of half-tablets than the scored furosemide tablets. Moreover, the weight of 4 out of 20 fragments that were split by patients deviated at least by 15%.

**Data Availability Statement:** All relevant data are within the manuscript and its Supporting information files.

**Funding:** The author(s) received no specific funding for this work.

**Competing interests:** The authors have declared that no competing interests exist.

## Conclusions

This finding showed that the tablet-splitting practices are poor and do not meet the specifications set by pharmacopoeias. Splitting by patients resulted in significantly higher dose variation and weight loss of fragments than splitting by pharmacists.

## Introduction

Tablet splitting is a practice where tablets of higher strength are broken in half, thirds, quarters, or into more pieces to provide the patient with the correct dose. To perform this practice with minimal errors it needs further expertise including selecting and identifying the candidate tablets for splitting. This exercise is considered *compounding* by the pharmacist because the prescription is customized to give the appropriate dosage and goes further than dispensing a commercially prepared product [1, 2]. Splitting scored tablets is approved by the FDA as efficacious and safe [3]. It is not considered problematic if the lowest available strength on the market is divided to provide a specific dose for a patient who needs a lower dose of a drug. However, according to the FDA Modernization Act of 1997, Section 503A, the pharmacist can violate the federal statute if a tablet is split to reproduce a dosage that is commercially available [3].

Tablet splitting and manipulation are widely practiced in many areas of healthcare. In Germany, in the primary care setting in 2006, an estimated one-quarter of all drugs were split [4]; and, in a large elderly care home in Canada, 35% of all tablets were split [5].

Several studies have been conducted to evaluate weight uniformity of split tablets. Elliott *et al.* investigated eight frequently split narrow therapeutic index or critical dosage medications for weight uniformity [6]. Of the eight studied medications, five failed to comply with the European Pharmacopoeia recommendations for half-tablet weight uniformity. The results showed a significant difference in splitting accuracy performed by nurses compared to laypersons (P = 0.027) which may potentially affect treatment outcomes.

Recently, Gharaibeh et al conducted a study in 491 patients to evaluate tablet-splitting practices in Jordan private pharmacies [7]. Twenty-four percent of respondents sometimes skipped their doses due to tablet splitting difficulties. The majority of participants (n = 312, 63.5%) used their hands to split tablets; while 29 (5.9%) participants used their teeth to split tablets. More than a tenth of the participants discarded parts of their tablets when splitting did not result in equal parts from their perspective.

In low-income countries, tablet splitting is even more commonly performed because of the limited range of dosage forms that are available and to minimize cost. However, the practice and implications have rarely been investigated. This results in splitting associated errors like weight variation, splitting tablets that are not candidate for splitting (for example, enteric-coated or sustained release tablets), and missing splitting and taking the whole tablets (overdose) [6]. Generally, the majority of tablet splitting studies were done in developed countries and many of them showed variations in weight/dose uniformity.

This study was aimed at determining the overall tablet splitting practices at Felegehiwot and Gondar University hospitals in Amhara Region, Northwest Ethiopia and providing ideas for potential interventions that would help improve these practices. Moreover, since there have not been similar studies in Ethiopia so far, it is expected that the findings of this study will serve as a foundation for further studies.

## Materials and methods

### Study area and setting

The study was conducted at two public hospitals in the Amhara Region of Ethiopia—Felegehiwot Referral Hospital and Gondar University Referral Hospital. Both hospitals were purposively selected for the study based on their high patient load, the suitable setup to easily access the required data, and their willingness to be enrolled in the study.

Feleghiwot Referral Hospital is located in Bahir Dar Town, Northwest Ethiopia which is the capital city of the Amhara Region. The hospital has around 400 beds and serves over 7 million people from the surrounding area. The second hospital, Gondar University Referral Hospital is a teaching hospital with a catchment area of around 6 million people.

### Study design and sampling strategy

A hybrid of study methods (prospective cross-sectional, observational, and experimental methods) were used to see the overall practices of tablet splitting in selected public hospitals in Ethiopia.

**Questionnaire study.** A prospective, cross-sectional study was conducted in outpatient pharmacies from September 2019 to December 2019. All pharmacy professionals who worked during the study period and have a willingness to participate were included in the study. Adult patients (18 years and older) who have been prescribed at least one medication in a split dosage form and attended the outpatient pharmacies during working hours were invited to participate in the study of tablet-splitting practices. Participants in both groups were selected based on their fulfilment of inclusion criteria for the study, and after providing informed consent to partake in the study. Separate questionnaires were developed for patients and pharmacists following an extensive literature review [2, 3, 6, 8, 9] and were checked by performing a pretest. It was performed by calculating 5% of the sample size for both patients and pharmacy professionals.

Data was also obtained through direct observation in dispensary units using observational tools. For example, presence of guidelines on tablet splitting, devices used for splitting, pharmacists' counselling and selection of candidate drugs, etc. were evaluated through direct observation.

### Sampling and sample size determination

A single proportion estimate was used to determine the sample size for patients in the quantitative part using the following formula. The prevalence of patients taking a split drug was taken from a previous study to be 17.5% [10].

$$n = \frac{(Z\alpha/2)2 \ p(1-p)}{d2} = \frac{(1.96)2(0.825)(0.175)}{(0.05)2} = 222$$

Where '$n$' is the minimum sample size required; '$Za/2$' is the critical value for a given confidence interval; '$\alpha$' is the probability of type I error; '$p$ (prevalence)' is proportion of patients with a prescription having a split drug; and '$d$' is absolute sampling error that can be tolerated.

Considering a 10% contingency, final sample size was calculated to be 222+22 = 244. However, during the survey, a total of 241 patients participated. On the other hand, all pharmacy professionals working in both hospitals at the time of data collection were included in the study using the census approach. A Pre-test was performed to validate the questionnaires.

### Inclusion and exclusion criteria of study participants

**For patients.** *Inclusion criteria*. Age 18 and above, willingness to participate in the study and prescribed at least one medication in a split dosage form were selected to be enrolled in this section of the study.

*Exclusion criteria*. Patients who are seriously ill and have a physical disability (for splitting only) and first time the patient had a prescription with a split tablet.

**For pharmacy professionals.** *Inclusion criteria*. All pharmacy professionals who work in the outpatient pharmacy during the study period and willingness to participate in the study.

*Exclusion criteria*. Who are seriously ill and have physical disability (for splitting only).

## Experimental part

### Weight uniformity and comparison of tablet splitting techniques

Two drug products of high potency that were commonly split were selected based on the data obtained from the study. Based on the above criteria, furosemide 40 mg (scored) and haloperidol 5 mg (unscored) was selected. According to the quality attributes of the tablets labelled as having a functional score in the USP, the test procedure involved taking 30 intact tablets from each product, weighing them out individually, and collectively using a sensitive (four decimal place) digital balance. The weight of the tablet half fragments was determined by dividing the theoretical weight of intact tablets by two since each tablet is split into two halves [11].

Six volunteers, three patients and three pharmacy professionals were recruited for splitting both the study products. Based on the findings of the tablet splitting practice, kitchen knives and bare hands were the most widely used techniques in the area. Accordingly, tablet splitting was done using a kitchen knife and bare hands to simulate the actual practice. Initially, the weights of the intact tablets were measured using a sensitive balance. Then, the weights of the half fragments were measured for each tablet. The split halves were weighed individually and then collectively to examine the effect of the splitting technique on the weight variation of tablet halves and the total loss from each tablet because of splitting. The average weight and standard deviation of the split halves were calculated and weight variation test was carried out according to the standards of the European Pharmacopoeia [12]. The criteria used in the study were adopted from the European Pharmacopoeia standards for the division of scored tablets which allows for no more than 1 in a set of 30 tablets to be outside the 85–115% range [12]. If one tablet falls outside this range, it must fall within 75–125% of the expected mass/content with RSD of less than 6%. Weight loss due to splitting should be less than 3%. Finally, the actual weights of split halves were compared to the weights of the theoretical perfect splits (t-test for independent means, $\alpha = 0.05$). Comparisons were made between the two sets of split tablets (split by patients vs. split by pharmacists) and the corresponding intact weights of the tablets before split (t-test for independent means, $\alpha = 0.05$) to determine the variation of means of fragment weights when using the two techniques (bare hands vs. kitchen knife).

### Content uniformity assay

The assay for both drugs was performed by preparing standard calibration curves. For validity and linearity of the method, 6 different concentrations (1μg/ml, 2μg/ml, 4μg/ml, 6μg/ml, 8μg/ml and 10μg/ml) were prepared by serial dilution from a standard working solution of the drug in 0.1 M sodium hydroxide for furosemide tablet. Then their absorbance were measured in triplicate in a quartz cuvette using a single beam UV/Vis spectrophotometer (MY15400018, Agilent Technology, Malaysia) at $\lambda_{max}$ of 275 nm. The unknown samples were prepared according to the British Pharmacopoeia (10) as follows: A number of 20 furosemide half-

tablets were randomly selected and crushed using mortar and pestle. The powdered quantity theoretically equivalent to 0.2 g of furosemide was added to 300 ml of 0.1 M sodium hydroxide in a 500 ml-volumetric flask and sonicated for 10 minutes. Then, a sufficient amount of 0.1 M sodium hydroxide was added to produce 500 ml and filtered using a filter paper. Five millilitres of the filtrate was further diluted to 250 ml with 0.1 M sodium hydroxide and its absorbance was measured at the same wavelength.

Similarly, for haloperidol, 20 fragments were randomly selected and crushed in mortar and pestle. Then, 10 mg of the powder theoretically equivalent to 0.5 mg of haloperidol was taken and transferred to a 40 ml volumetric flask and the volume was adjusted to its mark by using a diluent (methanol: water 80:20) and filtered with filter paper. Then, the sample was measured at $\lambda_{max}$ of 244 nm. The standard calibration curve was prepared by serial dilution of a stock solution of haloperidol working standard to six different concentrations (5μg/ml, 10μg/ml, 15μg/ml, 20μg/ml, 25μg/ml and 30μg/ml) using methanol: water (80:20) as a diluent and measuring the absorbance of the solutions at the same wavelength.

The final content uniformity for each fragment is obtained by cross-tabulation from their actual weight, assuming each tablet has uniform active ingredient contents. The repeatability of the test was assured by 6 determinations at 100% of test concentration.

## Statistical analysis

Data entry and analysis was done using SPSS version 20 statistical software and Microsoft excel. Descriptive statistics of the study population were done by analysing the distribution of the participants by the variables in terms of frequency and percentage. Statistical tests like Chi square test, t-Test for two independent data sets and F-test were used for data analysis and interpretation with P-values of less than or equal to 0.05 considered significant.

## Ethical consideration

Ethical clearance was obtained from the Ethics Review Board of the School of Pharmacy, College of Health Sciences, Addis Ababa University with an approval number of SoP 826/19. Written informed consent was obtained from the Medical Directors of the hospitals after the nature of the study, its purpose, and the potential benefits were described. Written informed consent was also obtained from all individuals participating in the study. They were informed that the information they provided would be kept confidential and analysed in aggregate.

## Data management and quality assurance

The questionnaire interviews for patients were administered by trained pharmacists. Pre-test (5% of the sample size) was performed and modification of the data collection tools was made before the commencement of the actual data collection. Data completeness was assured throughout the data collection, data entry and analysis by checking each questionnaire on the day of data collection. Validation of data entry was done by cross checking the hard copy with the database.

## Results

### Practice of tablet splitting

In this section of the study, a total of 241 patients and 82 pharmacy professionals participated. Separate questionnaires were prepared for the two groups. Table 1 shows the demographic distribution of participants. Additional data were also obtained through observation in

**Table 1. Distribution of patient and pharmacy professional participants involved in the study of tablet splitting practice.**

| Participants | Variables | Category | Frequency | Percentage |
|---|---|---|---|---|
| **Patients** | Sex | Male | 114 | 47.3% |
| | | Female | 127 | 52.7% |
| | Age (Years) | 18–28 | 46 | 19% |
| | | 29–38 | 76 | 31.5% |
| | | 39–48 | 69 | 28.6% |
| | | 49–59 | 39 | 16.2% |
| | | 59–69 | 11 | 4.7% |
| | Education Status | No formal education | 116 | 48.1% |
| | | Primary education (1–8 Grade) | 45 | 18.7% |
| | | Secondary education (19–12 grade) | 43 | 17.8% |
| | | College/ University | 37 | 15.4% |
| | Residency | Rural | 122 | 50.6% |
| | | Urban | 119 | 49.4% |
| | Religion | Christianity | 201 | 83.4% |
| | | Muslim | 40 | 16.6% |
| **Pharmacy** | Sex | Male | 52 | 63.4% |
| | | Female | 30 | 36.6% |
| | Education Status | Masters | 8 | 9.8% |
| | | Degree | 66 | 80.5% |
| | | Diploma (10 + 3 Grade) | 16 | 19.5% |
| | Work Experience | 0–2 years | 2 | 2.4% |
| | | 2–4 years | 4 | 4.9% |
| | | 4–6Years | 25 | 30.5% |
| | | >6years | 51 | 62.2% |

dispensary pharmacy units to explore the real practice of tablet splitting. More than half of patient participants (52.6%) were females.

**Tablet splitting practice of patients.** All participants were between 18 and 66 years old (average of 42 ± 13.2 years) and nearly half (48%) of them were illiterate. One hundred and twenty-one (50.6%) of the participants came from the surrounding rural areas (Table 1).

In all dispensary areas, there were no standard splitting devices, operating procedures (SOPs) and splitting areas in all pharmacy dispensary units for tablet splitting and manipulation. The task of splitting was totally given to patients or caregivers. Drugs, which are not candidates for splitting, were also dispensed to patients to be taken after a split. Enteric-coated formulations were dispensed to be split, despite the precaution given by the manufacturers against splitting or crushing the products for administration. For example, in Metoprolol 50 mg and Sodium Valproate 200 mg enteric-coated tablets, there is a precaution which reads as "*the drug should not be crushed, split or manipulated to be administered*" yet, these enteric-coated products were dispensed to be split into lower strengths. Some of the pharmacy professionals were not aware of the fact that these products are not candidates for splitting (see Table 2).

The majority of patient participants (51.3%) faced difficulties during splitting. There were no statistically significant differences with regard to age ($\chi2 = 4.5$; $p = 0.47$), but had a significant association with education level ($\chi2 = 60.5$; $p = 0.001$). Participants who attended higher education in general faced less difficulty during splitting compared to illiterate participants. Two problems were common, i.e., the tablets were hard and difficult to divide (10.3% of

**Table 2. List of non-candidate drug products (tablets) that were inappropriately split during the study period.**

| Drug product | Reason for non-candidacy |
|---|---|
| **Sodium valproate* 200mg** | Enteric coated |
| **Chlorpromazine* 100mg** | Enteric coated |
| **Metoprolol* 100 mg** | Enteric coated |
| **Acetyl salicylic acid* 300mg** | Enteric coated |
| **Furosemide*40mg** | Split into 8 fragments |
| **Digoxin[a] 0.25mg, warfarin[a] 5mg, benzhexol[a]5mg** | Potent |
| **Sulfamethoxazole and trimethoprine[a] (400mg + 80mg)** | Two active ingredients |

[a] = not well analysed/ insufficient data to restrict,

* totally restricted to be split (sufficient data)

The American Pharmacist Association recommends against splitting of film coated and unscored tablets, despite not included in this table [9].

respondents) and the tablets broke into several pieces or crumbled (18% of respondents). Because of this, 15 respondents (6.2%) believed that they encountered problems in controlling their health problems. Some chronic ambulatory patients said when they split tablets, sometimes the tablets break into unequal fragments and they discard them. Because of this, their medications are finished and they are forced to spend some days off medications until their next appointment dates to get a refill. Although 222 (92.3%) of patient participants responded they had received counselling, only 31(12.8%) patients were able to reiterate the instructions they got from pharmacists on how to split the tablets. Nearly half of the patients, 111 (46%), said they would prefer not to split the tablets; instead, they wanted appropriate strength whole tablets equivalent to their prescribed doses. In the case where tablet splitting resulted in unequal fragments, 22% of the participants (n = 53) stated that they waste parts of their medications and buy new medications from different sources, while 4% of participants (n = 10) stated they take the whole tablets without splitting. The rest 74% responded that they take the fragments with their limitations.

Twenty-one (8.7%) patients sometimes forgot to split their medications and take the whole tablets instead. Residence of the patients has a statistically significant effect on the extent of forgetting to split tablets (p = 0.013), the extent of forgetting being higher among rural participants than in urban ones. Twenty-three (9.5%) of respondents sometimes use their teeth to split drugs (Fig 1).

Education level has statistically significant association with the techniques used to split tablets (p = 0.001). Illiterate patients mostly use their hands or teeth, while educated patients mostly use kitchen knives or scissors for splitting.

During the study, a total of 279 tablet products that require splitting were dispensed to the 241 patient participants. Of these, 206 participants had only one tablet to split; 32 participants had 2 tablets to split; and 3 participants had 3 tablets to split. Incidentally, 53 (19%) of the 279 dispensed products were unscored. The most commonly split medication was phenytoin 100 mg (n = 32, 11.5%) followed by digoxin 0.25 mg (n = 24, 8.6%). Majority of the drugs were split into halves (n = 218, 78%); fifty-nine (21.1%) were split into quarters and, on two occasions, furosemide 40 mg was ordered to be split into 8 fragments. Moreover, split medications included enteric-coated and modified-release formulations such as metoprolol 100 mg, chlorpromazine 100 mg, aspirin 300 mg and sodium valproate 200 mg (n = 31, 11%). Incidentally, a scored captopril 25 mg tablet product was available in the dispensary units and the score line for this product was not at all symmetrical. Fortunately, during the study period, this product

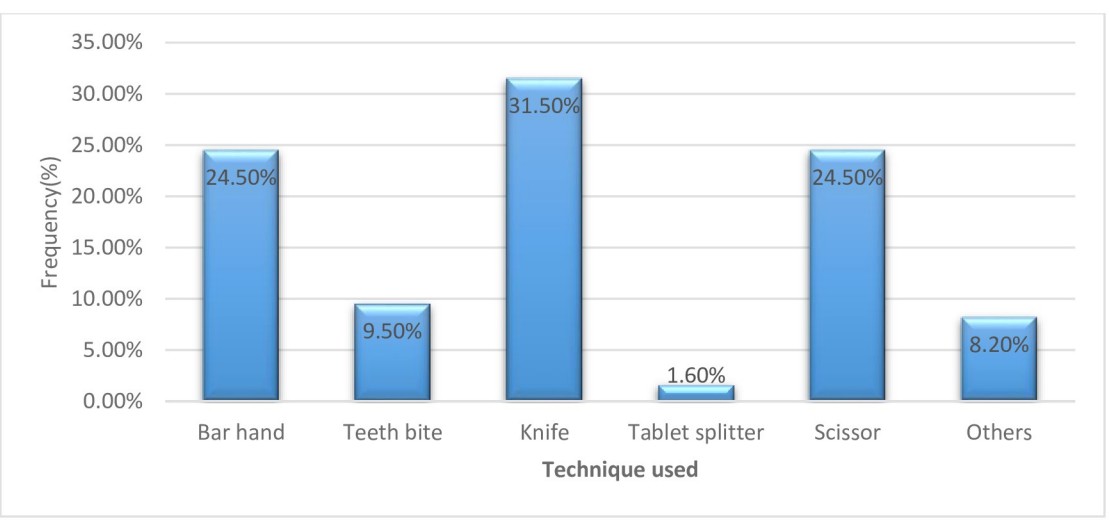

**Fig 1. Distribution of different tablet splitting techniques used by patient participants.**

was not given out to patients to take after splitting since a lower-strength dosage form was available.

**Tablet splitting practice of pharmacy professionals.** In this section of the study, a total of 82 pharmacy professionals participated, of which 16 were Diploma holders in pharmacy and 66 were degrees in pharmacy (Table 1). The majority of these participants were male, 52 (63.4%). Seventy-six (90%) of the participants had work experience of more than four years.

None of the respondents took training on tablet splitting and/or manipulation and only forty-five (55%) of the pharmacy professionals have hands-on experience in tablet splitting. Eighty percent of the pharmacy professionals responded that they give the task of splitting to patients or caregivers while the remaining 20% responded they counsel patients to look for the exact prescribed dose strength from private pharmacies instead of splitting higher dose products. When tablets with lower strength are available and the cost is nearly similar to the higher strength product, 41 (50%) of the respondents preferred to dispense the higher strength product to help patients save cost. Four of the participants indicated that they have occasions of forgetting to give instructions to patients to take their medications after splitting. The participants were also asked whether they have encountered drug therapy problems (toxicity or dose insufficiency) because of splitting problems. Accordingly, two had encountered overdoses with warfarin 5 mg and phenobarbital 100 mg. The first respondent encountered warfarin toxicity when he worked at Felegehiwot referral hospital in the paediatric ward as a clinical pharmacist. The dose prescribed to the patient was warfarin 2.5 mg. However, warfarin was available only in 5 mg dose strength. Apparently, the caregiver split the drug in two unequal parts and inadvertently gave the patient the larger fragments only. Because of this, the child developed warfarin toxicity and visited the hospital with a complaint of gum bleeding. The second respondent encountered phenobarbital overdose. The patient who was on phenobarbital therapy complained of excessive sleepiness. Then, the pharmacist assessed the problem and finally found the problem to be because the patient was taking a double dose of 100 mg, disregarding the instruction given by the pharmacist to split the tablets and take half fragments (50 mg).

Only 63.5% of pharmacy professional respondents selected enteric-coated products as non-candidates for splitting. The education level of the pharmacy professionals had a statistically

**Table 3. Pharmacy professionals' knowledge regarding non-candidacy of different tablet formulations for splitting.**

| Formulation type | Frequency | Percent |
|---|---|---|
| Multi-layered tablets containing more than one active ingredient | 16 | 19.5 |
| Enteric-coated tablets | 52 | 63.5 |
| Potent tablets (less than 25 mg) | 18 | 22 |
| Easily friable tablets | 26 | 31.7 |

significant association with identifying drugs that are candidates for splitting (p = 0.012). Six from 8 MSc holders (75%), forty from 58 BPharm pharmacists (69%) and only six from 16 druggists (37.5%) identified enteric-coated tablets as non-candidates for splitting. Only two participants knew the pharmacopoeial standards for accepting the weight uniformity of split tablets (see Table 3).

Enteric-coated, multi-layered, potent, unscored and easily friable tablets are not good candidates for splitting [9, 13]. However, the knowledge of the participants on the issue was not encouraging. Hence, intervention in the form of continuing professional development or other equivalent modalities of training is required. It is also interesting to note that 12% (n = 10) of pharmacy professional respondents believed conventional and scored tablets are not candidates for splitting.

## Weight and content uniformity of split tablets

**Weight uniformity of split tablets.** Furosemide intact tablet weights were within the allowed range with an average weight of 0.200 g and 3.15% RSD. The measured tablet weights expressed as a percent of target weight were found to be within the proxy USP specification percentage range (85%-115%). Upon splitting by the three patient participants using kitchen knives, the mean weights of fragments were reduced to 0.0964 g, 0.0970 g and 0.0966 g from 0.100 g theoretical weight with high variability (17.6%, 16.2% and 18.01% RSD, respectively). Nineteen (31.7%), fourteen (23.3%) and twenty (33.3%) out of 60 halves exceeded the allowed percent deviation of 15% from the mean for the 1st, 2nd and 3rd patient participants, respectively.

Furosemide intact tablets were also split by three pharmacists and the mean weights of the halves were 0.0980g, 0.0984 g and 0.0979 g with less variability (5.5%, 4.2%, 5.9% RSD) when compared with the halves split by patients. Unlike patient participants, only one fragment exceeded the allowed percent deviation when the first pharmacist participant split 30 tablets. All fragments were within the allowed USP specification when the second pharmacist split the tablet, while three fragments failed to meet the specification when split by the third pharmacist.

The loss in tablet weight as a result of splitting the tablets by the three patients using kitchen knife were 3.3%, 3.10% and 3.5%, which are higher than the 3% limit for weight loss of split tablets [13]. But in the case of pharmacist participants, the weight loss was 2.2%, 1.8% and 1.4% for the three participants, i.e., the weight loss was lower than the limit in all cases, unlike the patient participants.

The percentage distribution of half-tablet weights split by the patient and pharmacist participants around the labelled claim half-tablet weight is shown in Fig 2. As can be seen in the figure, the percent weight distributions of the intact furosemide tablets before being split by the patient and pharmacist were almost similar and closer to 100%. However, the distribution of the half-tablets split by the patient was more dispersed around the mean compared to the half-

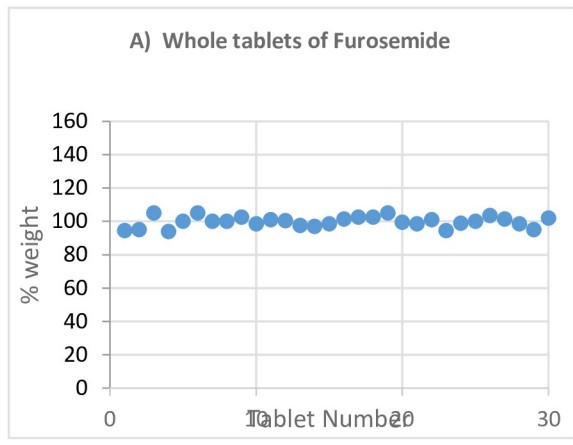

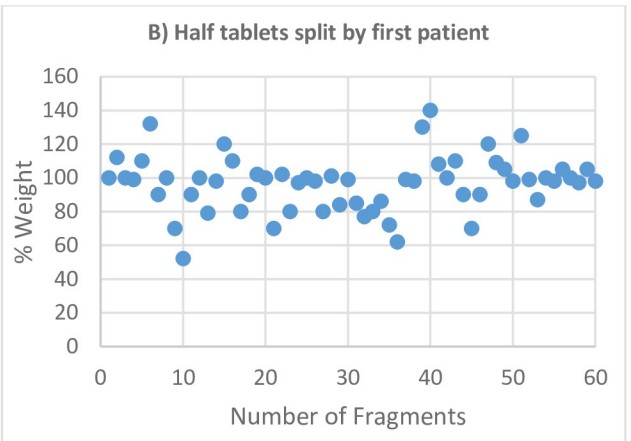

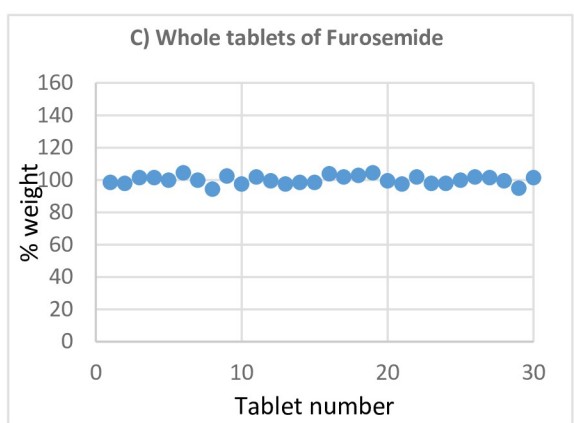

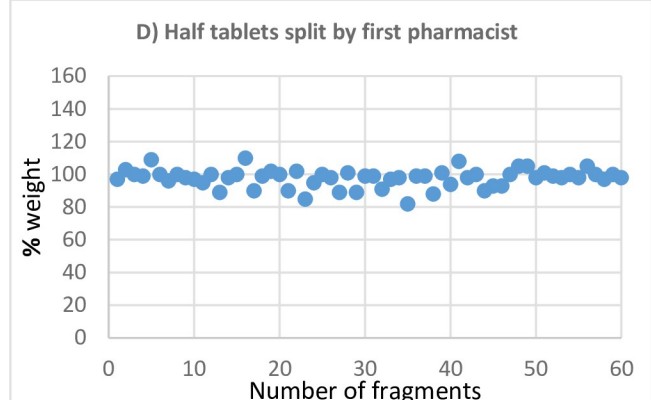

**Fig 2. Percentage weight variation of furosemide tablets from theoretical weight before and after split by patient and pharmacist participants.**

tables split by the pharmacist. The closer the distribution of weights of the half tablets around 100% the closer it is to perfect split (high accuracy).

The second drug in this study was haloperidol 5 mg, which had labelled claim weight for the intact tablet of 0.100 g including the excipients. Weights of the intact tablets were within the allowed range having an average weight of 0.100 g and 2.8% RSD. Accordingly, the theoretical weight for the half tablets was expected to be 0.050 g. However, upon splitting by three patient participants, the mean weights of the fragments were reduced to 0.048 g, 0.047 g and 0.046 g with RSD of 11.25%, 13.1% and 11.9%, respectively. Weight loss while splitting by the first patient was 0.004 g (4%). Moreover, sixteen (27%) out of 60 half fragments exceeded the allowed percent deviation of 15% from the mean. Similarly, splitting was done by the first pharmacist participant and the mean weight of fragments was similar to the result obtained for the patient participant (0.048 g). However, the RSD was reduced to 8.5% and ten (16.6%) out of 60 tablet fragments exceeded the allowed percent deviation.

The percentage distribution of half-tablet weights of haloperidol split by the patient and pharmacist participants around the labelled claim half-tablet weight is shown in Fig 3. As can be seen in the figure, the percent weight distributions of the intact furosemide tablets before split by the patient and pharmacist were almost similar and closer to 100% like furosemide

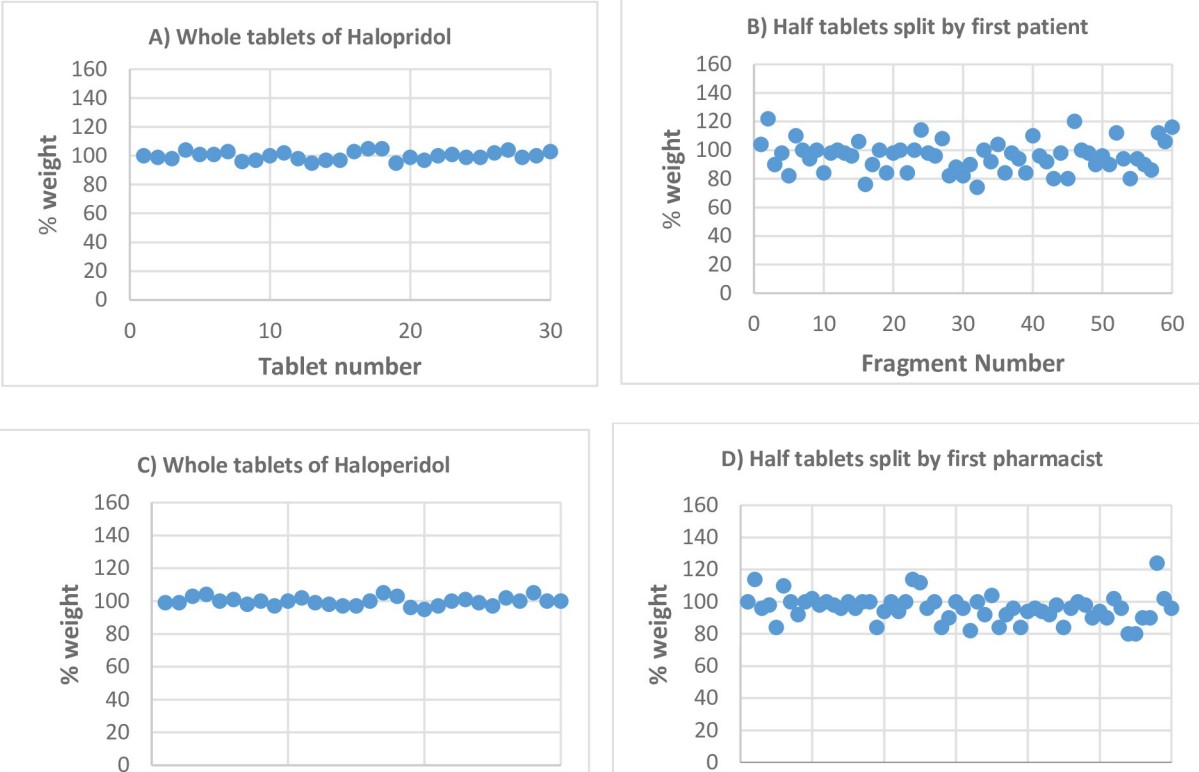

**Fig 3. Percentage weight variation of haloperidol tablets from theoretical weight before and after splitting by patient and pharmacist participants.**

tablets in the previous Fig 2. However, the distribution of the half-tablets split by both patient and pharmacist was more dispersed around the mean.

In many cases, during splitting, especially by patients using kitchen knives, drugs were split unsuccessfully into more than two pieces and crumbled (Fig 4). Therefore, this section of the study included only those splits, which successfully resulted in two fragments (halves) by rejecting the unsuccessful ones.

Similarly, splitting was done by patient and pharmacist participants using bare hands. The weight loss, as well as weight variation of the split fragments of the scored furosemide 40 mg tablets was significantly low when splitting was done by hands only, but splitting was very hard in the case of the unscored haloperidol tablets.

*Statistical analysis t-Test for two independent samples.* The aggregated mean weight of half tablets of furosemide split by patients using kitchen knife was 0.0967 g with SD of 0.0165, while the corresponding mean weight of half-tablets split by patients using bare hands was 0.0994 g with SD of 0.0059. The t-value calculated was 2.07 with a degree of freedom of 358; meanwhile, the t-value tabulated is 1.97 at 95% confidence level. The t-value calculated is higher than the tabulated t-value indicating that there was a statistically significant difference between the two means. Similarly, the aggregated mean weight of half tablets of furosemide split by pharmacists using kitchen knives was 0.0981 g with SD of 0.0047, while the corresponding mean weight of half tablets split by bare hands was 0.0999 g with SD of 0.0042. The

(A)                                                          (B)

**Fig 4. Observed weight loss (A) and weight variation (B) of split fragments.**

t-value calculated was 3.83 with a degree of freedom 358. The tabulated t-value was 1.97 which was lower than the calculated value at 95% confidence level. Hence, the mean weights have statistically significant differences when splitting by hand and using a kitchen knife.

F-test value for the two sets of data was analysed to compare the population variance. The standard deviation of furosemide tablets split by patients using kitchen knives was 0.0165 and the SD of tablets split by patients' bare hands was 0.0059. Accordingly, the F-value calculated was 7.82 with a degree of freedom of 179 for both sets of data; meanwhile, the F-value tabulated is 1.45 at 95% confidence level. Accordingly, the F-value calculated was more than the tabulated F-value and that gives evidence of unequal population variances. Similarly, the standard deviation of furosemide tablets split by pharmacists using kitchen knives was 0.0047 and the SD of tablets split by patients' bare hands was 0.0042. So, the calculated F- value was 1.26 with a degree of freedom 179. Accordingly, the F-value calculated was less than the tabulated F-value and that gives evidence of statistically insignificant difference between the population variances when splitting was done by pharmacist using kitchen knife or bare hands.

## Content uniformity of split tablets

The content uniformity of fragments split by patient and pharmacist participants of both tablet products was performed using UV spectrophotometry to see if the contents of the fragments meet pharmacopoeial specifications. Fig 5 shows the standard calibration curve of furosemide in 0.1N sodium hydroxide solution at a wavelength of 275 nm.

Similarly, the content of the furosemide tablets was determined after they were split using hand only. The mean absorbance of the unknown concentration was 0.068 and based on the calibration curve the concentration of furosemide was 1.6 µg/ml. As mentioned in the method, the dilution factor while sample preparation was 25000 (100 $^*$250). So that the concentration of the furosemide in the 0.200 g sample was 40 mg. Based on the cross tabulation, 1 out of 20 fragments which were split by patient deviated at least by 15% whereas the content of the 20 fragments split by pharmacist participants were within the acceptable 85%-115% range.

Fig 6 shows the standard calibration curve of haloperidol in methanol: water (80:20) medium at a wavelength of 244 nm.

The mean absorbance of the sample solution was 0.560, which corresponds to a concentration of 12.2 µg/ml. Considering a dilution factor of 40, the concentration in a total 10 mg

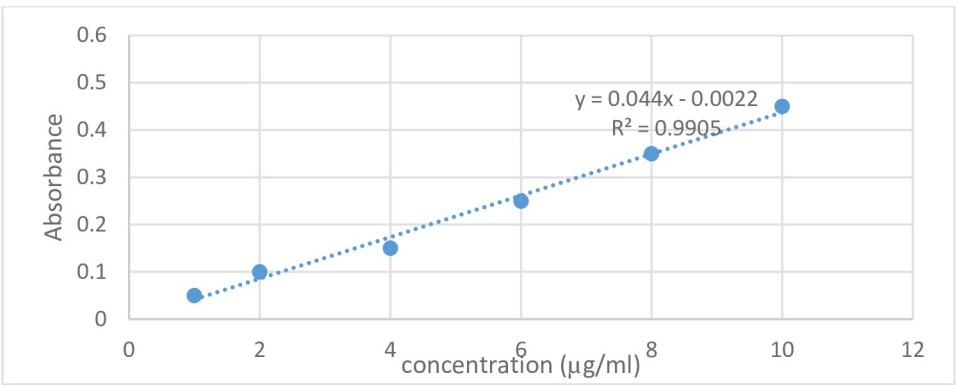

**Fig 5. Standard calibration curve of furosemide at wavelength of 275 nm.** The mean absorbance of the unknown concentration was 0.070 and based on the calibration curve, the concentration of the sample is 1.62 μg/ml. The dilution factor while sample preparation was 25000 (100 *250). Hence, the amount of active ingredient in the total sample (0.2g) is 40.5 mg. Based on the finding on UV/Vis spectrophotometry at 275 nm wavelength, 0.2 g sample contained 40.5 mg furosemide. So, the amount of each fragment can be cross tabulated assuming each intact tablet had a uniform furosemide amount. Based on the cross-tabulation, when the tablets were split using a kitchen knife, 4 out of 20 fragments which were split by patient deviated at least by 15% whereas the content of all 20 fragments split by the pharmacist participants were within the 85%-115% range.

powder was 488 μg haloperidol. The cross-tabulation results for each fragment showed that 6 out of 20 fragments of haloperidol which were split by patient and 4 out of 20 fragments split by pharmacist deviated by at least 15% from their label claimed amount of half tablets (2.5 mg).

The content uniformity test of both drugs did not meet the pharmacopoeial standards when the tablets were split by patients; in the case of furosemide, the content uniformity met the EP specifications when split by pharmacist.

## Discussion

Tablet splitting is widely practiced throughout the world, but has rarely been examined in low-income countries. Frequency of splitting tablets is expected to be higher in developing countries due to the availability of a limited range of dosage forms. However, a significant share of

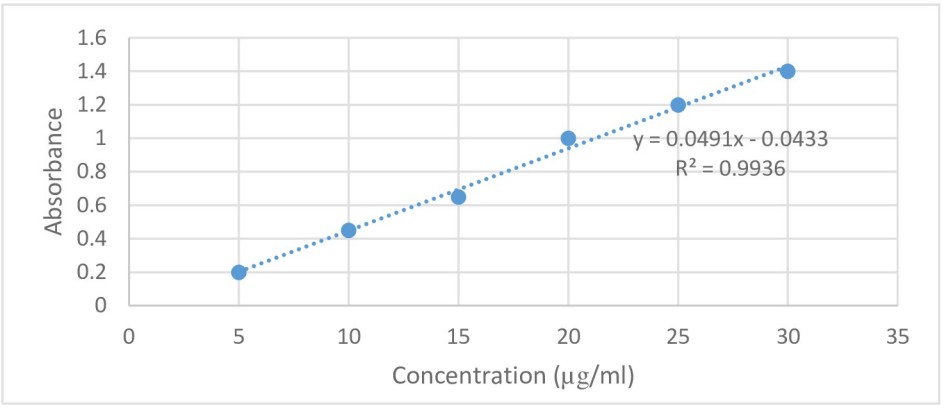

**Fig 6. Standard calibration curve of haloperidol at wavelength of 244 nm.**

patients have trouble for dividing. This and many previous studies have shown patients encounter problems like non-uniform splitting when they have to take fragments of their medications [9, 14–16].

The majority of medications intended for splitting in this study were drugs for the central nervous system. A study with the same design was done in Sweden in 346 patients and nearly one-third of patients (31.4%) faced difficulties while splitting [15]. Similarly, a study in Jordan reported around one-fourth (24%) of participants sometimes skipped their doses due to tablet-splitting difficulties [7]. The number is much larger in the current study with 51.3% of the participants facing difficulties during splitting. The reason for this is probably patients in the present study did not have access to standard splitting devices as compared to 37.2% and 42.2% of participants had access in the studies by Anders and Ekedal and by Gharaibeh et al, respectively [7, 15].

A significant number (9.5%) of patients split tablets by biting them apart with their teeth in this study, compared to only 4.4% in Anders and Ekedahl [7]. This technique obviously results in non-uniform fragments and can be of clinical significance in cases of narrow therapeutic index medications. Clearly, this practice has to be discouraged during patient counselling. The devices they should use, the purpose of score line on the tablets, the purpose of splitting, what they should do if splitting results in unequal fragments and so on were not properly addressed to the patients [17–19].

Incidentally, 53 (19%) of the 279 dispensed products to be taken after split were unscored, which is much higher than a study done at a primary care setting in Germany where only 66 (8.7%) of 762 drugs were unscored [4]. The most commonly split medication in this study was phenytoin 100 mg (n = 32, 11.5%) followed by digoxin 0.25 mg (n = 24, 8.6%). Both medicines have a narrow therapeutic index, and hence higher risk for suboptimal treatment or adverse effects. On the contrary, aspirin 325 mg was most commonly split medication (n = 187, 38.1%) followed by digoxin 0.25 mg (n = 16, 3.3%) in the study by Shadi *et al.* in Jordan [7]. In the current study, 78% of the drugs were split into halves, where as a study in Brazil by Vivienne *et al.* found that 96% of the tablets were split in halves [8].

The impact of tablet splitting on patient adherence to prescribed drugs was evaluated by Choe *et al.* and Stankovic *et al.* and found that a significant number of patients forgot splitting and instead took the whole tablet [20, 21]. Similar result was obtained in the current study with 8.7% of patients forgetting to split their medications. Denneboom *et al.* found that problems to divide tablets were the second most common clinically relevant risk factors for non-adherence among elderly patients. Accordingly, pharmacists should be encouraged to ask patients how patients manage to adhere to their prescribed treatments [22]. Extended release dosage forms and enteric coated tablets like metoprolol and sodium valproate were freely dispensed to be split, although the instruction in the labels of these products clearly indicates not to do so. Enteric coated dosage forms and unscored tablets accounted for 11% and 19%, respectively of all the products which were dispensed to the study patients to be taken after splitting.

Splitting by many participants failed to meet the EP standards on weight uniformity of divided fragments. Previous studies reported similar findings [6, 23–25]. Verrue *et al.* split eight drugs using three mostly used techniques in Belgium: splitter devices, hand split and kitchen knife as a result only one drug met the USP/EP standards. These findings are broadly similar to the current study, despite the use of different tablets and techniques of splitting [26].

Splitting of scored furosemide tablets by hand resulted in evenly split fragments as well as insignificant weight loss when compared to splitting with a kitchen knife. A study by Diana *et al.* in the Netherlands compared the accuracy of splitting using three techniques: hand breakage, kitchen knife and splitter devices and the result was similar with the current study

in that hand broken tablets were more uniform than tablets split by kitchen knife [27]. Similarly, in a study by Elliot *et al.* splitting of tablets by hand resulted in more accurate splits than using a scissor. Tablet parts or fragments were tested to see whether they complied with three regulatory requirements adopted to the conditions of the experiment: Ph. Eur. subdivision of tablets; assay; and FDA loss of mass. Only hand broken, scored tablets complied with all three tests and the weight loss was significantly low. Crumbles and powder parts were clearly seen on the surface of the knife during splitting and that was the reason the weight loss was higher [6, 28].

Recently, a study by Vivienne *et al.*, in Brazil showed the mean mass loss was 8.7% after split using their preferred splitting techniques, which is much higher than the findings of this study and a study reported by Teixeira *et al.*($<$2%). This study shows unscored tablets are too difficult and results in non- uniform fragments. The content uniformity test of both drugs did not meet the pharmacopoeial standards when the tablets were split by patients [8, 29].

In Ethiopia, splitting devices are not widely available and splitting is mostly done either by bare hand or using a kitchen knife. On the other hand, pre-splitting assessment by pharmacists is not done before dispensing the split tablets. Due to these reasons, splitting errors are expected to be much higher than in high-income countries. As a general conclusion, based on the findings of the current study, in the absence of a splitting device, hand splitting could be a better solution than kitchen knife splitting for scored tablets.

This study however has limitations like, the study was done only on available drugs during the study period. Hence, some tablets were out of stock and were available after few days and this affected the assessment of the frequency of splitting of those drugs. If they were available throughout the collection period, the frequency of splitting might have been higher than this finding.

## Conclusions

This finding showed that the tablet-splitting practices are poor and do not meet the specifications set by pharmacopoeias. Splitting by patients resulted in significantly higher dose variation and weight loss of fragments than splitting by pharmacists. Furthermore, splitting by hand resulted in more uniform fragments than splitting by kitchen knives. To the best of our knowledge, no available guidelines regulate the tablet-splitting practice in Ethiopia.

### Recommendations

Appropriate interventions should be initiated by providing adequate training to pharmacists and other healthcare professionals about tablet splitting and manipulation (both pre-service and in-service training).

It is also important to establish nationwide comprehensive guidelines and standards for tablet splitting and manipulation practises. This initiative can be led by the competent regulatory authority of the country, the Ethiopian FDA. In addition, the government could support the public pharmaceuticals import and distributor, Pharmaceutical Supply Agency, as well as the private sectors to import a wide range of paediatric dosage forms and standard splitting devices to decrease the splitting frequency as well as splitting errors. At institutional level, detailed guidelines and standard operating procedures could be developed to guide these practises in a more streamlined and structured fashion. Local pharmaceutical manufacturers should be also supported to produce a wide range of dosage forms.

Pharmacists should evaluate a person's ability to break a tablet accurately and determine the most suitable method of breaking.

## Supporting information

**S1 Data.**
(PDF)

## Acknowledgments

The authors want to thank Addis Ababa University for providing ethical approval to the study
and the selected hospitals for their positive cooperation during the study. We would also like
to forward our gratitude to the study participants.

## Author Contributions

**Conceptualization:** Asmamaw Emagn Kasahun, Anteneh Belete.

**Data curation:** Asmamaw Emagn Kasahun.

**Methodology:** Asmamaw Emagn Kasahun, Nisha Mary Joseph, Anteneh Belete.

**Supervision:** Nisha Mary Joseph, Anteneh Belete.

**Validation:** Anteneh Belete.

**Visualization:** Anteneh Belete.

**Writing – original draft:** Asmamaw Emagn Kasahun.

**Writing – review & editing:** Asmamaw Emagn Kasahun, Nisha Mary Joseph, Anteneh Belete.

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
