## [Decision Letter · Decision Letter 0]

11 Oct 2022

PONE-D-22-15692Tablet dosage form splitting practices and dose uniformity of fragments at public hospitals in Ethiopia: a Mixed, cross sectional and experimental study methodsPLOS ONE

Dear Dr. Belete,

Thank you for submitting your manuscript to PLOS ONE. After careful consideration, we feel that it has merit but does not fully meet PLOS ONE’s publication criteria as it currently stands. Therefore, we invite you to submit a revised version of the manuscript that addresses the points raised during the review process.

We look forward to receiving your revised manuscript.

Kind regards,

Amjad Khan, Pharm-D; PhD

Academic Editor

PLOS ONE

“The authors would like to recognize the Addis Ababa University and for its support and smoothing the study.

Reviewers' comments:

Reviewer's Responses to Questions

**Comments to the Author**

1. Is the manuscript technically sound, and do the data support the conclusions?

Reviewer #1: Yes

Reviewer #2: Partly

2. Has the statistical analysis been performed appropriately and rigorously? 

Reviewer #1: Yes

Reviewer #2: Yes

3. Have the authors made all data underlying the findings in their manuscript fully available?

Reviewer #1: Yes

Reviewer #2: Yes

4. Is the manuscript presented in an intelligible fashion and written in standard English?

Reviewer #1: Yes

Reviewer #2: No

5. Review Comments to the Author

Reviewer #1: The manuscript gives sound information about research methodology and research question. The study is well organized however the topic is vast more drugs can be selected for better comparison. The author can give the reason of choosing the specific drugs for the study. On what basis these two hospitals are selected? Questionnaire distributed in pharmacy and hospitals doesnot include similar questions.

Reviewer #2: An excellent effort to improve pharmacy practice in Ethiopia. Some suggestion are given to improve the manuscript

1. English need extensive revision.

2. Abstract and conclusion should be re-written according to results

3. Method:

A. Please enlist the studies which were considered to develop questionnaire?

B. What were the results of reliability testing?

C. How the validity was monitored.?

D. How the same questionnaire justifies the healthcare workers and patients ?

E. How the participant were selected?

F. What were the rational to exclusion and inclusion criteria ?

6. PLOS authors have the option to publish the peer review history of their article (what does this mean?). If published, this will include your full peer review and any attached files.

Reviewer #1: No

Reviewer #2: **Yes: **Muhammad Majid Aziz

---

## [Author Response · Author response to Decision Letter 0]

10 Nov 2022

Response to Reviewers 

Tablet dosage form splitting practices and dose uniformity of fragments at public hospitals in Ethiopia: a Mixed cross-sectional and experimental study methods

The authors would like to thank reviewers for their insightful, constructive comments. In the preparation of this response, the Reviewer’s comments have been numbered as indicated in the appended annotated copies of the Reviewer’s evaluations and supporting files.

Response to Editors 

We note that you have provided funding information that is not currently declared in your Funding Statement. However, funding information should not appear in the Acknowledgments section or other areas of your manuscript. We will only publish funding information present in the Funding Statement section of the online submission form. Please remove any funding-related text from the manuscript and let us know how you would like to update your Funding Statement. Currently, your Funding Statement reads as follows: “The author(s) received no specific funding for this work.”

The authors acknowledge the editor’s comment. The authors did not receive funding from Addis Ababa University. However, the university supports us by providing ethical clearance to undertake the study. We modified the acknowledge part accordingly. 

Response to reviewer 1

The authors appreciated the reviewer’s comments

The study is well organized however the topic is vast more drugs can be selected for better comparison. The author can give the reason of choosing the specific drugs for the study.

The authors accepted the reviewer’s comment and modified the title accordingly. The authors select two drugs (Furosemide and haloperidol) because of their high potency, were commonly split based on the data obtained from the primary cross-sectional study and were available during the study period.

On what basis these two hospitals are selected?

Both hospitals were purposively selected for the study based on their high patient flow, their willingness to be enrolled in the study and the setup of pharmacy units were suitable and found near to access information easily. 

Questionnaire distributed in pharmacy and hospitals does not include similar questions.

The aim of the cross-sectional study section is not to compare the knowledge of pharmacy professionals with patients. Hence, the contents of the questionnaires are different for patients and pharmacists.

Response to reviewer 2

The authors appreciated the reviewer’s comments

1. English need extensive revision.

Thank you, we revised accordingly.

2. Abstract and conclusion should be re-written according to results

 Thank you, revised accordingly.

3. Method:

A. Please enlist the studies which were considered to develop questionnaire?

 The authors accepted the reviewer’s comment 

B. What were the results of reliability testing and C) How the validity was monitored?

The authors accepted the reviewer’s comment. We included methods we followed to assure the reproducibility and validity of our test. “For validity and linearity of the method, 6 different concentrations (1μg/ml, 2μg/ml, 4μg/ml, 6μg/ml, 8μg/ml and 10μg/ml) were prepared by serial dilution from a standard working solution of the drug in 0.1 M sodium hydroxide for furosemide tablet.” The repeatability of the test was assured by 6 determinations at 100% of test concentration. 

 D. How the same questionnaire justifies the healthcare workers and patients?

 The contents of the questionnaire were different for Pharmacists and patients. 

 E. How the participant were selected? And What were the rational to exclusion and inclusion criteria ?

Thank you for your comments. The authors modified the exclusion and inclusion criteria. The participants were selected with clearly justified inclusion and exclusion criteria for both cross-sectional and experimental studies.

---

## [Decision Letter · Decision Letter 1]

1 Dec 2022

Practices of Tablet Splitting and Dose Uniformity of Fragments at Public Hospitals in Ethiopia: a cross-sectional study supported by experimental findings

PONE-D-22-15692R1

Dear Dr. Belete,

We’re pleased to inform you that your manuscript has been judged scientifically suitable for publication and will be formally accepted for publication once it meets all outstanding technical requirements.

Kind regards,

Amjad Khan, Pharm-D; PhD

Academic Editor

PLOS ONE

Additional Editor Comments (optional):

Reviewers' comments:

Reviewer's Responses to Questions

**Comments to the Author**

1. If the authors have adequately addressed your comments raised in a previous round of review and you feel that this manuscript is now acceptable for publication, you may indicate that here to bypass the “Comments to the Author” section, enter your conflict of interest statement in the “Confidential to Editor” section, and submit your "Accept" recommendation.

Reviewer #1: (No Response)

2. Is the manuscript technically sound, and do the data support the conclusions?

Reviewer #1: Yes

3. Has the statistical analysis been performed appropriately and rigorously? 

Reviewer #1: Yes

4. Have the authors made all data underlying the findings in their manuscript fully available?

Reviewer #1: Yes

5. Is the manuscript presented in an intelligible fashion and written in standard English?

Reviewer #1: Yes

6. Review Comments to the Author

Reviewer #1: The author has addressed the comments and made necessary changes in the paper. It's recommended to accept the paper.

7. PLOS authors have the option to publish the peer review history of their article (what does this mean?). If published, this will include your full peer review and any attached files.

Reviewer #1: No

---

## [Editor Report · Acceptance letter]

4 Dec 2022

PONE-D-22-15692R1 

Practices of Tablet Splitting and Dose Uniformity of Fragments at Public Hospitals in Ethiopia: a cross-sectional study supported by experimental findings 

Dear Dr. Belete:

I'm pleased to inform you that your manuscript has been deemed suitable for publication in PLOS ONE. Congratulations! Your manuscript is now with our production department. 

Kind regards, 

on behalf of

Dr. Amjad Khan 

Academic Editor

PLOS ONE